# Synthesis of Naphthalene-Based Polyaminal-Linked Porous Polymers for Highly Effective Uptake of CO_2_ and Heavy Metals

**DOI:** 10.3390/polym14061136

**Published:** 2022-03-11

**Authors:** Manal Ibrahim, Nada Tashkandi, Nikos Hadjichristidis, Nazeeha S. Alkayal

**Affiliations:** 1Chemistry Department, Faculty of Science, King Abdulaziz University, BOX 80203, Jeddah 21589, Saudi Arabia; maliibrahim@stu.kau.edu.sa (M.I.); nytashkandi@kau.edu.sa (N.T.); 2Physical Sciences and Engineering Division, Polymer Synthesis Laboratory, KAUST Catalysis Center, King Abdullah University of Science and Technology (KAUST), Thuwal 23955, Saudi Arabia; nikolaos.hadjichristidis@kaust.edu.sa

**Keywords:** polyaminal-linked polymers, naphthaldehyde, melamine, CO_2_ uptake, heavy metals removal

## Abstract

Studying the effect of functional groups on the porosity structure and adsorption efficiency of polymer materials is becoming increasingly interesting. In this work, a novel porous polyaminal-linked polymer, based on naphthalene and melamine (PAN-NA) building blocks, was successfully synthesized by a one-pot polycondensation method, and used as an adsorbent for both CO_2_ and heavy metals. Fourier transform infrared spectroscopy, solid-state ^13^ C NMR, powder X-ray diffraction, and thermogravimetry were used to characterize the prepared polymer. The porous material structure was established by field-emission scanning electron microscope and N_2_ adsorption–desorption methods at 77 K. The polymer exhibited excellent uptake of CO_2_, 133 mg/g at 273 K and 1 bar. In addition, the adsorption behavior of PAN-NA for different metal cations, including Pb(II), Cr(III), Cu(II), Cd(II), Ni(II), and Ba(II), was investigated; a significant adsorption selectivity toward the Pb(II) cation was detected. The influence of pH, adsorbent dose, initial concentrations, and contact time was also assessed. Our results prove that the introduction of naphthalene in the polymer network improves the porosity and, thus, CO_2_ adsorption, as well as the adsorption of heavy metals.

## 1. Introduction

Sources of energy (coal, oil, and natural gas) have had a substantial impact on human society’s progress. On a worldwide scale, fossil fuels are regarded as the primary energy source; nevertheless, anthropogenic emissions of carbon dioxide from these sources raise CO_2_ levels far above the natural carbon cycle, causing various environmental problems and, in particular, the greenhouse effect [1,2].

Another environmental problem arises from increased industrial activity affecting aquatic environments due to the amounts of heavy metals being discharged and added to their extreme water solubility. They integrate into the food chain, causing harm to human health and the environment [1]. Therefore, activities like mining, smelting [3], fuel combustion [4], paint and dye production, waste treatment, and battery manufacturing, release wastewater, with a significant amount of heavy metals, into water resources [1].

Many methods have been attempted to solve these two issues. For example, adsorption, absorption, membrane treatment, and cryogenic distillation have been used for CO_2_ capture [5]. Precipitation, solvent extraction [3], ultrafiltration [4], and adsorption were employed to remove heavy metals. Among these methods, adsorption is one of the best choices and has proven to be inexpensive and practical, owing to its easy and cheap process. However, it has attracted many concerns in the past years. Many porous adsorbents such as mesoporous silica, graphenes, carbon nanotubes, magnetic particles, metal–organic frameworks (MOFs), biopolymers, and porous organic polymers (POPs) have been designed/synthesized for these purposes [6].

Porous organic polymers (POPs), with their flexibility of a synthesis approach, inherent porosity, low density, large specific surface areas, and high thermal and hydrolytic stability, have attracted much attention due to their different applications in gas uptake and separation, sensing [7], electronics [8], heterogeneous catalysis, environmental therapy, among others. Polyaminal networks (PANs) are a category of POPs that are of special interest in terms of their high N-content, rich microporous frameworks, and easy fabrication by a catalyst-free one-pot polycondensation reaction of aldehyde derivatives with amines [7]. Generally, the interaction between the amine and aldehyde group yields an imine linkage (–C=N–). However, when the amine group is bonded with a strong electron releasing group, the enhanced basicity enables it to interact with the developed imine bond, resulting in the formation of a stable aminal linkage (–NH–C–NH–) [9]. The synthesis of PANs has been realized through many efforts and synthetic strategies. In order to synthesize PANs with stable porous structures, rigid aromatic moieties have been used to prevent network shrinkage [10].

From the point of view of actual application, it is desirable that the initial raw materials for PANs be commercially available to enable simple and cost-effective fabrication. Melamine (MA) is an inexpensive chemical substance, containing 66% N by mass, and is commonly employed in the coating and plastic industry. In addition, melamine has considerably superior basicity, resulting from the electron-donating effect of the triazine ring on the primary amino groups [11]. It can react with other molecules through hydrogen bonding, metal coordination, and π-π interactions. As a result, it is an excellent candidate to be used as a starting substance to synthesize N-rich polymers [12].

Monofunctional benzaldehyde and its derivatives have been studied extensively in PANs fabrication. Recently, Adeela Rehman and her co-worker have synthesized PANs from benzaldehyde and p-toulaldehyde with a surface area of 30 and 33.5 m^2^/g, respectively, and abundant macro- and mesopores [2]. Furthermore, Biao Zhang and his group have synthesized a series of PANs from melamine with benzaldehyde and its derivatives (p-carboxylbenzaldehyde, p-nitrobenzaldehyde, and p-hydroxylbenzaldehyde). They observed that the BET surface area of unsubstituted PAN-P clearly increased by –OH and –COOH functionalization, but the pore size of the modified polymers shifted from 0.8 nm toward the mesoporous region [13]. In addition to the surface chemistry, it is well known that surface and pore properties are very important for adsorption applications. The pore size should be less than 0.8 nm for effective CO_2_ adsorption performance [14]. Therefore, it is preferable to build new PANs with improved pore characteristics. However, creating novel mono-aldehyde derivatives to obtain tailor-made functionalized PANs remains fascinating and highly demanding [10].

Polycyclic aromatic compounds with two or more fused benzene rings are an important family of organic molecules. They are beneficially rich in electrons ascribed to the π-conjugated structures [15]. Thus, they are possibly a better option for fabricating PANs with outstanding applications. Furthermore, naphthalene, which has a wide range of resources, is a significant polycyclic molecule when used to synthesize specific compounds. Nevertheless, as far as we know, the naphthalene group, with its rich active sites and electrons [14], has not received much consideration regarding the fabrication of porous polymers for the adsorption of both CO_2_ and metal cations. Therefore, the introduction of naphthalene into PANs is supposed to considerably enhance the porosity and adsorption behavior [16]. Furthermore, unlike the planar and rigid benzene, naphthalene can move from the flat environment and modify its topology. This flexibility affects the activity of the naphthalene ring, yet it has proven to be useful in the development of 3D architectures and the investigation of their characteristics [17].

Based on the previous consideration, in this study, α-naphthaldehyde as an electron-rich building block was selected for the fabrication of novel polyaminal-linked networks. Thus, it is possible to study the impact of naphthalene in the construction of microporous polymers, in the chemistry and structure of pores, and, therefore, on the adsorption capacities. Naphthalene-based polyaminal networks were prepared via a direct one-pot polycondensation of the monomers without using any catalyst (Figure 1). The structures and properties of the fabricated polymers were analyzed by Fourier transform infrared spectroscopy, solid-state ^13^C NMR, powder X-ray diffraction, and thermogravimetric analysis. The porous material structure was established by field-emission scanning electron microscope and N_2_ adsorption–desorption methods at 77 K. In addition, the potential applications of this polymer particle on the uptake of CO_2_ and heavy metals have been explored.

## 2. Materials and Methods

### 2.1. Materials

α-Naphthaldehyde (90–95%) was purchased from Fluka Chemie AG, Buchs, Switzerland; melamine (97.5%) and dimethyl sulfoxide (DMSO 99%) were supplied from BDH Laboratory Reagents, England, UK; tetrahydrofuran (THF ≥ 99.5%), ethanol (99.99%), dichloromethane (≥99.8%), and Pd (NO_3_)_2_ (98%) were purchased from Fisher Chemicals, England, UK; methanol (99.5%) was purchased from NTL; Cd(NO_3_)_2_·4H_2_O (99%) was purchased from Sigma–Aldrich, Darmstadt, Germany; NaOH (98%), CuCl_2_·2H_2_O, and NiCl_2_·6H_2_O (98% and 97%) were supplied by BDH Chemicals, England, UK; Ba(NO_3_)_2_ (99%) was provided by Ward’s Natural Science, Rochester, NY, USA; Cr(NO_3_)_3_·9H_2_ O (97%) was provided by Montplet & Esteban sa, Barcelona, Spain; and HCl (35%) was supplied from LOBA Chemie, Mumbai, India. All substances were used without additional purifications.

### 2.2. Synthesis of Naphthalene-Based Polyaminal-Linked Porous Polymer (PAN-NA)

A dry three-necked flask equipped with a magnetic stirrer and condenser was first evacuated with a vacuum and then degassed with an evacuation–argon-backfill cycle. Then, 0.5 g (3.96 mmol) of melamine and 0.69 mL (7.16 mmol) of α-naphthaldehyde were added to 25 mL of DMSO and heated to 175 °C for three days. Finally, the mixture was cooled down to room temperature and the product was collected by filtration and washed consecutively with additional methanol, THF, and dichloromethane to afford a yield of 85%.

### 2.3. Metals Adsorption

To establish the metal adsorption capacity of the polymer, a standard solution of Cr(III), Cu(II), Cd(II), Pb(II), Ni(II), and Ba(II) (20 mg/L) was prepared by mixing the salts of the metals with deionized water. A total of 100 mL of the solution was transferred into a beaker with 20 mg of PAN-NA and stirred for one hour. The mixture was then filtered and the filtrate was collected to measure the concentrations of the metals. The adsorption capacity q_e_ (mg/g) of PAN-NA and its removal efficiency R(%) were determined according to the following equations:$$q_{e}=\frac{V\;}{m}\;\left(C_{e}\;-\;C_{f}\right)$$
$$R\%=\frac{C_{e}-C_{f}}{C_{e}}\times 100$$
where C_e_ and C_f_ are the initial and final concentrations of metal cations (mg/L), respectively. V(L) is the solution volume and m(g) is the adsorbent amount.

#### 2.3.1. pH Effect

A standard solution containing 20 mg/L Pb(II) was prepared at 3, 4, and 6 pH values. A total of 1 M HCl and 1 M NaOH were used to adjust the pH. PAN-NA microporous polymer (20 mg) in a set amount and 100 mL of Pb(II) solution were mixed for 1 h at each pH value.

#### 2.3.2. Dose Effect

To examine the influence of adsorbent dose on the Pb(II) removal. The amounts of 10, 20, 30, and 40 mg of PAN-NA polymer were weighed and added to 100 mL of 20 mg/L Pb(II) solution at the optimum pH.

#### 2.3.3. Reusability of PAN-NA

In the desorption tests, 50 mL (0.5 M) of HCl solution and 50 mL of ethanol were utilized as the desorption medium to regenerate the adsorbent from the Pb(II)@PAN-NA complex. After shaking for 3 h at 298 K, the sample was filtered, washed with deionized water, and recycled using the adsorption–desorption process. For the test, the adsorption–desorption cycle was repeated 5 times [18].

### 2.4. Characterization Methods

A Perkin–Elmer spectrophotometer was used to record Fourier transform infrared FT-IR spectra in a wavenumber region of 400–4000 cm^−1^. The sample was prepared using the KBr disk method. A Bruker AVANCE I HD spectrophotometer (Thuwal, Saudi Arabia)was applied to record solid-state ^13^ C cross-polarization magic-angle spinning nuclear magnetic resonance (^13^ C CP/MAS NMR) spectra at 500 MHz. The sample morphology was obtained by a field-emission scanning electron microscope (FE-SEM, PhotoMetrics, Inc., Huntington Beach, CA, USA). An X’Pert PRO MPD diffractometer (Malvern Panalytical, Malvern, UK) was used to measure powder X-ray diffractions (PXRD) from 15° to 80°. Thermogravimetric analysis (TGA) was performed on a TG-DTA6300 (Shimadzu, Kyoto, Japan) with a 10 °C min^−1^ heating rate in an interval of 25–500 °C under N_2_ atmosphere. N_2_ adsorption–desorption measurements were conducted on a Micromeritics 3 Flex 3500. The Brunauer–Emmett–Teller (BET) and Langmuir methods were employed to calculate the surface area of the material. The t-plot was used to approximate the micropore surface area. Before analysis, the sample was degassed by heating at 120 °C for 12 h under vacuum. Non-local density functional theory (NLDFT) was utilized to determine the pore size distribution (PSD). CO_2_ adsorptions were measured at 273 K up to 1 bar. The solutions’ metal contents were determined using inductively coupled plasma optical emission spectroscopy (ICP-AES) on a Perkin–Elmer Optima 7000 DV (PerkinElmer, Inc., Waltham, MA, USA).

## 3. Results and Discussion

### 3.1. Synthesis and Characterization

Polyaminal PAN-NA (white powder) was successfully prepared by one-pot polycondensation in DMSO, as illustrated in Figure 1. PAN-NA is hyper-cross-linked, as evidenced by its total insolubility in water and many widely used organic solvents, including dimethyl sulfoxide, dichloromethane, tetrahydrofuran, and methanol. This feature allows it to be stable and, thus, to resist a wide range of organic solvent environments [9].

The structure of PAN-NA was investigated by FT-IR, solid-state ^13^C NMR, SEM, XRD, TGA, and surface area analysis. The FT-IR spectra (Figure 1) reveal the formation of aminal linkages in PAN-NA by the appearance of bands at 3365 and 1188 cm^−1^, assigned to the stretching and bending vibrations of the secondary amine (N–H). The intensity of the characteristic bands for –NH_2_ stretching and –NH_2_ bending at 3470, 3420 and 1650 cm^–1^, respectively, are considerably decreased when melamine units are formed [19]. New peaks appear at 1347 cm^–1^ and 2922 cm^–1^, assigned to the (C–N) and methylene (C–H) groups, respectively, of the aminal, the characteristic triazine ring bands are observed at 1541 and 1474 cm^–1^, confirming the existence of melamine groups in the structure [20]. Furthermore, the aldehyde group’s distinctive band can be seen, with little intensity, at 1678 cm^–1^, indicating that certain aldehyde moieties are not involved in the polymer networks. In general, the FT-IR spectra show that the melamine-based polymer formed with a high extent of polymerization.

To further verify the structure, solid-state ^13^C NMR analysis is recorded in Figure 2. The strong signal at 166 ppm was ascribed to the triazine ring carbon, and the broad signal at 53.9 ppm was assigned to the methylene group in aminal linkages. According to the FT-IR study result, no signals from unreacted aldehyde are identified in the ^13^C NMR spectra of PAN-NA. FT-IR technique is considerably more sensitive to carbonyl groups and thus may detect them at very minimal concentrations. Others have made related observations on such networks [21,22].

In Figure 3, the powder X-ray diffraction (PXRD) pattern of PAN-NA reveals a wide peak, positioned around 2θ = 20°, corresponding to the most amorphous part [5,23]. The lack of any sharp peaks may indicate the absence of melamine, meaning that all melamine has been reacted.

The field-emission scanning electron microscope (FE-SEM) technique is utilized here to examine the surface morphology of the polymer at a microscopic level (Figure 4). PAN-NA has a cotton-like shape with irregular tiny particles [2,24].

The thermal stability of PAN-NA material was explored by TG analysis and the thermogram is presented in Figure 5. While heating in N_2_ atmosphere, a weight loss of 5% was noticed up to 145 °C. This may result from the out-gassing of solvents and moisture stuck in the pores of the polymer network. The main weight loss began at a temperature of over 320 °C, due to the breakdown of the network, and the burning began gradually above 400 °C. Our results confirm that PAN-NA is thermally stable up to 320 °C.

Porosity characteristics are critical parameters influencing the adsorption effectiveness of small molecules on PAN surfaces. In addition, N_2_ adsorption–desorption isotherms were used to investigate surface areas and porosity characteristics at 77 K. As shown in Figure 6a, the abrupt growth in N_2_ uptake at the initial relative pressure (P/P_0_ < 0.01) suggest the presence of extensive micropore materials. Then, in the intermediate pressure range, nearly horizontal adsorption occurs. Furthermore, when relative pressure (P/P_0_) surpasses 0.8, N_2_ uptake increases dramatically, indicating the presence of bigger mesopores, which are mostly caused by interparticle voids induced by loose packing of fine particles, as demonstrated by SEM micrographs (Figure 4) [23,25,26]. On the other hand, the polymer exhibits reversible adsorption–desorption isotherms, which differ from the hysteresis phenomena commonly reported in many other porous organic polymers due to pore structural deformation during measurements in liquid N_2_. As a result of the reversible isotherms, network architectures built with triazine rings and aminal linkages are robust [11].

The pore-size distribution (PSD) of the polyaminal was evaluated by non-local density functional theory (NLDFT) (Figure 6b). Notably, the pore’s diameter is primarily centered in the ultra-micropore region at 0.54 nm. Several micropores at 1.27 nm also existed, showing that the produced PAN-NAs are ultra-microporous materials [24]. PAN-NA has a BET surface area of 607.46 m²/g and a t-plot micropore area (S_micro_) of 378.9 m²/g. The micropore contribution ratio to the overall specific surface area ratio (S_micro_/S_BET_) is 0.62, suggesting that PAN-NA belongs to microporous polymers. The micropore volume and total pore volume of PAN-NA are 0.153 cm³/g and 1.07 cm³/g, respectively.

### 3.2. CO_2_ Adsorption

The adsorption isotherm of CO_2_ is presented in Figure 7. Interestingly, CO_2_ uptakes increased consistently with increasing pressure and stayed below saturation throughout the experimental range of pressure, implying that a much higher adsorption capacity may be reached at higher CO_2_ supply pressures. At 273 K and 1 bar, PAN-NA shows a higher CO_2_ uptake of 67.9 cm^3^/g (133.32 mg/g), which is larger than the adsorption capacities of other porous polymers [13,27,28,29,30].

Generally, the polar groups have an excellent affinity to CO_2_ [16]. In PAN-NA polymers, the amine groups originating from melamine exhibit good attraction for CO_2_ molecules [5]. The possible reasons for the enhancement of the CO_2_ uptake of nitrogen atoms (from triazene and aminal linkage) are due to quadrupolar interaction, H-bond, and acid-base attraction via an interaction between the lone pair of N atoms with the partially positively charged C atom of CO_2_ [C(δ+)···N(δ–)] [31]. The higher amount of CO_2_ adsorbed could be also attributed to the microporosity of the polymer network [26].

### 3.3. Metal Adsorption

The adsorption capacities (mg/g) of the PAN-NA polymer toward metal cations are summarized in Table 1. PAN-NA exhibits a very good uptake to all cations, with a superior selectivity for the Pb(II) cation, owing to the freely available functional groups, the intrinsic microporosity, and the greater specific surface area, which results in a higher capacity for cationic species [1,32]. Thus, the Pb(II) cation was chosen for further adsorption experiments.

#### 3.3.1. Effect of pH and Adsorbent Dose

The solution pH is a critical variable in the adsorption system. The solution pH can influence the charge of the adsorbent surfaces, ionization degree, and adsorbent speciation [4]. Figure 8a clearly shows that the pH of the solution had a considerable impact on the Pb(II) adsorption efficiency. When the solution pH reached three, the excess of H^+^ in the solution protonated the nitrogen atoms in PAN-NA, leading the polymer surface to be positively charged [33]. As a result, the electrostatic repulsion decreases the adsorption capacity between adsorbent and Pb(II) cation [18]. As the pH increases, the nitrogen atom is deprotonated, and the polymer surface charge becomes increasingly negative.

Therefore, the adsorption capabilities of Pb(II) were raised significantly, owing to electrostatic attractions between ions with different charges [34]. When the pH exceeds six, however, –OH interacts with the Pb(II) cation, leading to the production of PbOH^+^ and Pb(OH)_2_, according to the lead distribution diagram [35]. Under these conditions, the mechanism of Pb(II) removal becomes more convoluted and distinguishing between the precipitation and adsorption of Pb(II) separate from the solution becomes difficult [18]. Thus, at pH = 6 the existing H^+^ for competing with Pb(II) decreased, allowing the adsorption efficiency to rise. The maximum adsorption capability of 90.36 mg/g was found at this pH level, which was then employed in subsequent adsorption studies. The removal effectiveness of Pb(II) increased with the increasing adsorbent dose, owing to additional active sites being accessible at higher amounts of the adsorbent. However, once the adsorption procedure reaches a saturation point, no additional Pb(II) cations can be adsorbed onto the polymer, regardless of adsorbent dosage [18]. The adsorbent dose of 20 mg was efficient for the adsorption experiment (Figure 8b).

#### 3.3.2. Adsorption Isotherms

The effect of initial concentration on the adsorption efficiency of PAN-NA was studied in the range of 10–200 mg/L. As shown in Figure 9a, the adsorption capacity of PAN-NA increased with an increase in the amount of Pb(II) cations in the solution to reach a maximum value of q_e_ = 970.75 mg/g at 200 mg/L. As we are interested in studying the adsorption capacity of PAN-NA at low concentrations, we choose the 20 mg/L for optimum adsorption conditions. Two isotherm models (Langmuir and Freundlich) were applied to the experimental data to understand the adsorption mechanism of Pb(II) on the polymer surface. The Langmuir isotherm model describes the monolayered adsorption on a homogenous surface [36]. The linear equation of the Langmuir model is used here as follows:$$\frac{C_{e}}{q_{e}}=\frac{C_{e}}{Q_{m}}+\frac{1}{{\mathrm{bQ}}_{m}}$$
where q_e_ (mg/g) and C_e_ (mg/L) are the adsorption capacity of polymer and Pb(II) concentration, respectively. b (L/mg) is the Langmuir constant and Q_m_ (mg/g) is the maximum adsorption capacity. The Freundlich isotherm model describes the multilayer adsorption on a heterogeneous surface [36]. The linear equation of the Freundlich model is used here as follows:$$\log q_{e}=\log k_{F}+\frac{1}{n}\log C_{e}$$
where k_F_ ana 1/n are Freundlich constants which describe the adsorption intensity and capacity. As shown in Figure 9b, it is clear that the Langmuir isotherm does not fit the adsorption data (R^2^ = 0.745). PAN-NA has a multilayer coverage of Pb(II) cations and the PAN-NA/Pb(II) interaction is heterogeneous, as illustrated by fitting well with the Freundlich isotherm model (R^2^ = 0.999) (Figure 9c). All Langmuir and Freundlich parameters are summarized in Table 2.

#### 3.3.3. Kinetic Study

The rate of the adsorption process is a crucial variable when selecting an adsorbent. The kinetics of Pb(II) adsorption on PAN-NA were studied over a time of 20–120 min. Figure 10a depicts the impact of contact time on Pb(II) adsorption capacity. The Pb(II) uptake reaches equilibrium within the first 20 min, suggesting the rapid Pb(II) adsorption efficiency of the PAN-NA polymer. Pseudo-1st-order and pseudo-2nd-order kinetic models were used to evaluate the experimental data (Figure 10b,c). Both kinetic models were applied using the following equations:

pseudo-1st-order kinetic model.
$$\log\left(q_{e}-q_{t}\right)=\log\left(q_{e}\right)-\frac{K_{1}t}{2.303}$$

pseudo-2nd-order kinetic model.
$$\frac{t}{q_{t}}=\frac{t}{q_{e}}+\frac{1}{K_{2}q_{e}^{2}}$$
where K_1_ (min^−1^) and K_2_ (g/mg·min) are the rate constants. q_t_ and q_e_ (mg/g) are the adsorption capacities at time t and equilibrium, respectively. According to the correlation coefficient (R^2^ = 1), the experimental data fits well with the pseudo-2nd-order kinetic model, implying that the adsorption process of Pb(II) on the PAN-NA surface takes place through chemisorption mechanism [3]. In addition, the calculated q_e_ value from the pseudo-2nd-order model is exactly equal to the experimental value (q_e_ = 100.7 mg/g). The pseudo-1st-order and pseudo-2nd-order constants are summarized in Table 3.

#### 3.3.4. Reusability and Recyclability

Adsorbent recycling for repeated applications is critical for achieving the most cost-effective treatment method [34]. As presented in Figure 11, after conducting the adsorption and desorption processes five times, we found that the polymer can be reused a maximum of two times without losing its efficiency. The adsorption capacities after two recycling runs at optimum conditions were 90.36% and 88.17%, respectively.

The current study found that PAN-NA is constructed very effectively with excellent yield and exhibited good chemical and thermal stability. By comparing our PSD result with analogs’ polyaminal with benzene moiety, we observed that PAN-NA has a smaller pore size than the benzyl polymer, which may be due to the fact that naphthalene groups occupy extra network space than benzene groups [2,9,13,27].

Therefore, we noticed that the carbon dioxide adsorption in the naphthyl polymer is higher than the benzyl polymer [2,27]. This may be due to two factors. First, CO_2_ uptake is pore-size dependent. The pore size has to be close to the CO_2_ diameter (3.30 Å) which leads to the strengthening of the adsorbed molecule and pore wall interaction, which increases the ability for CO_2_ uptake to occur at a low pressure with a certain pore volume [25]. Since naphthyl has double the active sites of benzyl, as a result, it will favorably enhance the cross-linking degree in the networks, allowing polymer segments to subdivide the space into tinier pores [14]. Second, the higher distribution of negative charge all over the building units has a significant impact in increasing CO_2_ adsorption ability, based on previous theoretical and experimental studies [14,37]. Due to the intrinsic strength of π-conjugated systems, naphthyl is beneficial for improving CO_2_ uptake [14].

The observed significant Pb(II) adsorption capacity by the PAN-NA polymer is an outstanding phenomenon which may be attributed to the following points: (i) amine groups can bond with the Pb(II) cation very efficiently as they act as coordinating ligands [1]; (ii) the addition of unique naphthalene spacer units, which may weaken hydrogen bonding and, therefore, activate the several amine and triazine groups [12]; and (iii) the particular surface shape and random structures with inherent microporosity, which is extremely advantageous for heavy metal ion adsorption [12]. Compared with prior works on Pb(II) adsorption, for instance, the porous melamine–vanillin polymer (MVP) (8.53 mg/g) [29] and the cross-linked melamine–pyridine polyaminal network (MA-Py) (53.13 mg/g) [1], our PAN-NA shows higher Pb(II) removal due to the aforementioned points.

## 4. Conclusions

In conclusion, a new polyaminal-based porous polymeric network from melamine and naphthaldehyde was fabricated by a one-pot polycondensation. The production and porous properties of the novel prepared polymer were explored. The PAN-NA showed good thermal stability up to 320 °C. It was found that the PAN-NA polymer possesses a large surface area of 604 m^2^/g and pore sizes of 0.54 and 1.27 nm, which revealed that this polymer belongs to ultra-microporous materials. The CO_2_ capacity of the adsorbent was determined at 273 K and 1 bar. PAN-NA had a CO_2_ uptake value of 133.32 mg/g. In addition, the correlation between the characteristics of the polymer structure and gas uptake behavior was evaluated and it was concluded that naphthyl, with additional active sites and increasing π-surface area features, is an excellent choice for the formation of microporous polymers with varied porosity and good performance of gas adsorption. The ability of the polymer in heavy metals adsorption was investigated. We conclude that the PAN-NA is selective toward the Pb(II) cation with an adsorption capacity of 100.7 mg/g at pH = 6, 20 mg/L, and an adsorbent dose of 20 mg at the end of 20 min. Our polymer was prepared from cheap raw materials in a facile synthetic approach with remarkable results in adsorbing CO_2_ and heavy metals. Thus, it is an excellent candidate for real environmental applications, such as selective CO_2_ separation from the gas mixture.

According to our findings, the introduction of naphthalene into the polymer improves its properties for different applications. Thus, it opens the area to the synthesis of such polymer using naphthaldehyde derivatives, and the study of the effect of functional groups on enhancing CO_2_ capture and heavy metals adsorption.

## Data Availability

Not applicable.

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
