# Peer review of "Synthesis of Naphthalene-Based Polyaminal-Linked Porous Polymers for Highly Effective Uptake of CO2 and Heavy Metals"

_polymers, 2022, doi:10.3390/polym14061136_

Round 1

Reviewer 1 Report

In this paper, the authors obtained naphthalene-based polyaminal-linked porous polymers for uptake of CO2 and various heavy metals, such as: Pb(II), Cr(III),  Cu(II), Cd(II), Ni(II), and Ba(II).

Regarding the removal of heavy metals, I suggest to include in the manuscript the following:

  1. Impact of initial Pb(II) concentration and isotherm models (i.e. Langmuir, Freundlich)
  2. Impact of contact time and kinetic models (i.e. Pseudo first order model, Pseudo second order model)
  • Why the experiments were carried out only for 1 h contact time? 1 h of contact time is sufficient in order to reach the equilibrium?

Introduction section

Lines 80, 91 -- CO2 replace with CO2

Line 95 -- Remove the space between ‘’….yet it has’’  and ‘’been proven…’’

Line 98 – Please verify: ‘’ novel polyaminal-inked networks’’

Section 2.1

  • Line 113 - 116

Pd(NO3)2 replace with Pd(NO3)2

Cd(NO3)2.4H2O replace with Cd(NO3)2 . 4H2O

CuCl2.2H2O replace with CuCl2 . H2O

NiCl2.6H2O replace with NiCl2 . 6H2O

Ba(NO3)2  replace with Ba(NO3)2

Cr(NO3)3.9H2O replace with Cr(NO3)3 . 9H2O

Section 2.3.2

  • Line 145 – Please correct the following paragraph: ‘’ 10, 20m 30, and 40 mg of PAN-NA polymer was weighted and added to 100 ml of 20 ppm Pb (II) solution 146 at the optimum pH’’

Section 3.1.

  • Please verify the notation: Scheme 1 or Scheme I?
  • The appearance of Figure 2 is not suitable at all
  • Line 187 – Please replace ‘’1650 Cm–1’’ with ‘’1650 cm–1’’

Section 3.2.

  • Line 263 – Please replace ‘’ 9 cm3/g’’ with ‘’ 67.9 cm3/g’’

Section 3.3.

  • The graph of the impact of adsorbent dose on the Pb (II) adsorption should be included
  • Line 301 – Please remove the point between ‘’ adsorbent’’ and ‘’but’’
  • Line 327 – Please verify the paragraph: ‘’ (ii) (ii) The higher distribution of negative charge all over building units per- 327 forms a significant impact in increasing CO2 adsorption ability, based on previous theoretical and experimental studies [14,33]’’
  • Line 340 – Remove the space between ‘’…due to’’ and ‘’the above mentioned points’’.

Conclusions section

  • Line 346 – Please replace ‘’604 m2/g’’ with ‘’604 m2/g’’

References section

References 17, 18, 30, 31, and 32 are list in ‘’References section’’, but are not included in the manuscript body.

Author Response

Dear Ms. Huang

First, the authors thank the reviewer for these comments that enhance the quality of the manuscript.

Please find attached our revised manuscript. polymers-1623145 entitled “Synthesis of naphthalene-based polyaminal-linked porous polymers for highly effective uptake of CO2 and heavy” by Alkayal et al.

We have revised the manuscript according to the reviewers’ comments (changes in the MS are marked with yellow). Please find all referees’ comments as well as our point-by-point responses.

Best regards

Nazeeha Alkayal

Reviewer 2 Report

This is an interesting research work dealing with the synthesis of naphthalene-based polyaminal-linked porous polymers for the adsorption of  CO2 and heavy metals, such as Pb(II), Cr(III), Cu(II), Cd(II), Ni(II), and Ba(II). The paper is well structured and interesting findings have been found by the researchers. The paper can be further processed in the journal after some moderate revisions:

1) Please, double check the grammar of the entire paper along with punctuation.

2) Please, update your introduction section with the ongoing progress on novel nanocomposite membranes for the separation of heavy metals from contaminated water. The new readers need to know the current state of the art in the field.

3) The authors need to declare the main issues when adapting this new polymer into membranes. What do they expect about the applicability? Please, give feedback on it.

4) the authors may also give feedback about the possible adsorption mechanism and interaction of the polymer-molecule uptake.

5) Similar to point 5, please when stating the current advances in composite membranes for heavy metal removal, make a comparison with other studies in the field. we should always make a comparison with other studies' results.

6) In the conclusion section: It is important to declare what is next? Any perspective of this new polymer? maybe adapting it as a membrane for real application in membrane processes, e.g. CO2 separation, Separation and Purification Technology 254 (2021), 117582, solvent separation via pervaporation,  water treatment, among others.

Author Response

(The authors gave the same response as above.)

Reviewer 3 Report

The paper entitled “Synthesis of naphthalene-based polyaminal-linked porous pol-2 ymers for highly effective uptake of CO2 and heavy metals” by Manal Ibrahim, Nada Tashkandi, Nikos Hadjichristidis and Nazeeha S. Alkayal submitted for publication in Polymers achieves a new polymeric material, with porous capacity, based on naphthalene and melamine. Thanks to the porosity it can trap carbon dioxide or heavy metals. In particular there is a good selectivity towards Pb(II).

In the Materials and Methods the subindexes should be included for all the compounds, take for instance Cd(NO3)2.4H2O.

Chemically, apart from the selectivity towards heavy metals, did the authors check the selectivity on different gases? The CO2 adsorption should be compared to past studies, take for instance with MOFs (J. Am. Chem. Soc. 2013, 135, 15986−1598, DOI: 10.1021/ja407135k; Inorg. Chem. 2018, 57, 6981-6990, DOI: 10.1021/acs.inorgchem.8b00670…)

The references should be checked, take for instance in reference 15 “CO 2” should be corrected. But mainly this section should be improved since it is neither complete nor updated. Actually, just 32 citations seem to be a rather low number, and the missing comparison with other materials, the selectivity of gases, and also the capacity of adsorption of metal ions.

Author Response

(The authors gave the same response as above.)

Round 2

Reviewer 1 Report

Authors have made the necessary corrections. Therefore, I recommend the publication of the article.